# Development of *Cytisus* Flower Extracts with Antioxidant and Anti-Inflammatory Properties for Nutraceutical and Food Uses

**DOI:** 10.3390/ijms26157100

**Published:** 2025-07-23

**Authors:** Adela Alvaredo-López-Vizcaíno, Augusto Costa-Barbosa, Paula Sampaio, Pablo G. del Río, Claudia Botelho, Pedro Ferreira-Santos

**Affiliations:** 1Department of Chemical Engineering, Faculty of Science, University of Vigo (Campus Ourense), As Lagoas, 32004 Ourense, Spain; adela.alvaredo@uvigo.gal (A.A.-L.-V.);; 2Instituto de Agroecoloxía e Alimentación (IAA), University of Vigo (Campus Auga), As Lagoas, 32004 Ourense, Spain; 3Centre of Molecular and Environmental Biology (CBMA), Universidade do Minho, Campus de Gualtar, 4710-057 Braga, Portugal; augusto.ac.barbosa@gmail.com (A.C.-B.); psampaio@bio.uminho.pt (P.S.); 4Institute of Science and Innovation for Sustainability (IB-S), Universidade do Minho, Campus de Gualtar, 4710-057 Braga, Portugal; 5CEB—Centre of Biological Engineering, University of Minho, Campus de Gualtar, 4710-057 Braga, Portugal; claudiabotelho@ceb.uminho.pt; 6LABBELS—Associate Laboratory, 4710-057 Braga/Guimarães, Portugal

**Keywords:** green extraction, broom flowers, phenolic compounds, antioxidant activity, anti-inflammatory properties, sustainable valorization

## Abstract

Plant flowers are recognized as a rich source of bioactive phenolic compounds. In this study, for the first time, the recovery of antioxidant phenolic compounds from *Cytisus striatus* flowers (CF) was optimized using microwave-assisted extraction (MAE). The variables (% of ethanol, temperature, and time) were studied using a response surface methodology (RSM). Extraction efficiency was assessed by total phenol content, total flavonoid content, and the antioxidant capacity through DPPH, ABTS, FRAP, and CUPRAC assays. Additionally, cytotoxicity and anti-inflammatory properties were evaluated in different cell lines. The optimal extraction conditions (87.6% ethanol, 160.8 °C and 8.76 min) yielded extracts rich in phenolics (85.9 mg GAE/g CF) and flavonoids (120.3 mg RE/g CF), with strong antioxidant capacity. LC-MS/MS analysis identified 27 phenolic compounds, including chrysin, apigenin, and quercetin derivatives. Cytotoxicity tests showed that CF extract maintained high viability (>80%) in human embryonic kidney (HEK293T) and human lung adenocarcinoma (A549) cells up to 2000 µg/mL, indicating low cytotoxicity. The anti-inflammatory potential was evidenced by a decrease in IL-1β levels and an increase in IL-10 cytokine production in LPS-stimulated macrophages. These results highlight the great potential of CF as a promising bioresource to obtain value-added compounds for the development of functional foods, nutraceuticals, and cosmetic products.

## 1. Introduction

The world’s population has increased from 3.7 billion to 8 billion people in the last 50 years, and a UN estimate is an increase to 9.8 billion by 2050 [1]. This rapid population increase has intensified the demand for effective food production systems to satisfy growing nutritional needs. However, this increase in food production has been accompanied by significant challenges, including increased health problems related to the quality of diets and environmental degradation, as well as the production of large quantities of by-products and discarded “residues”. The by-products generated, such as agri-food and forestry discards, despite being increasingly utilized complying with the demands of the European and national authorities (Circular Economy and Waste Management in alignment with United Nations Sustainable Development Goals-2030 (SDGs)), many of them often remain underutilized, leading to pollution and waste accumulation [2].

To meet these challenges, it is imperative to implement sustainable management practices that focus on the valorization of these by-products. According to European directives, these agri-food and forestry by-products can be directly used as fertilizers, compost, or biofuels, subject to certain safety conditions and industrial viability. In addition, through advanced biorefinery approaches, these by-products can be harnessed for their potential nutritional value and/or transformed into demanded high-value products such as bioactive compounds, functional ingredients, and enhanced ecological agricultural inputs [3,4,5,6,7,8]. These practices not only reduce environmental impacts but also increase the value and opportunity of reusing and recycling by-products, contributing to a circular bioeconomy. This integrated approach has the potential to increase the population’s needs while promoting health, sustainability, and resource efficiency [9].

The plants of the genus *Cytisus*, better known as brooms, belong to the family Fabaceae and include about 70 species. These plants are distributed all over the world, being prevalent in open sites of the Mediterranean Basin (Europe and North Africa) and western Asia. Brooms are normally undervalued plants and considered as invasive or ornamental species. Evidence of their traditional use for infusions and decoctions due to their interesting source of bioactive compounds, like polyphenols and alkaloids, with relevant therapeutic properties [10,11,12,13,14,15].

The extraction methodology is a crucial process for the effective recovery of bioactive compounds. These techniques must be designed to be adaptable, cost-effective, eco-friendly, and simple to use on an industrial scale. They also need to ensure high extraction yields and preserve the quality of the biocompounds extracted [16]. The range of state-of-the-art extraction technologies includes a multitude of innovative methodologies like ultrasounds, electrotechnologies, etc., and microwave-assisted extraction (MAE) ranks prominently for the recovery of bioactive compounds [15,17,18].

MAE is an innovative and advanced technique that uses microwave energy to accelerate and improve the efficiency of extraction of bioactive compounds from various matrices [18,19,20]. MAE energy interacts with the molecular dipoles and ions present in the solvent and biomass, generating heat rapidly and uniformly. This rapid, localized heating causes cell walls to rupture, facilitating the release of bioactive compounds [18]. MAE, compared to other techniques, can reduce extraction time and minimize solvent use, which makes it attractive for numerous industrial applications. However, it requires careful optimization of extraction parameters such as time, temperature, and solvent selection specific to the target compounds from selected bioresources [21]. Response surface methodology (RSM), together with a design of experiments (DOE), is a sophisticated statistical approach for the optimization of extraction processes, determining the best combination of tested parameters (independent variables) to maximize the intended result (dependent variables), improving extraction efficiency [22]. The RSM statistical approach has been associated with MAE in numerous studies to improve the efficiency of extraction of bioactive compounds from food and agricultural by-products, reducing resource consumption and processing time for the extraction of the compounds of interest [22].

The objective of the present study was to optimize the extraction of bioactive phenolic compounds from the *Cytisus striatus* flowers (Figure 1) using RSM. The resulting phenolic-rich extract was chemically characterized by HPLC-TOF-MS, and its cytotoxicity, antioxidant, and anti-inflammatory activities were evaluated. These findings support the potential application of *C. striatus* extracts in nutraceutical and functional food formulations.

## 2. Results and Discussion

### 2.1. Chemical Characterization of Cytisus Striatus Flowers

The chemical composition of *C. striatus* flowers used in this work is presented in Table 1. This data reveals a distinctive composition with notable elements (nutrients and bioactive compounds) with imperative potential application in food, pharmaceutical and cosmetic industries, and agriculture sector. After drying at 37 °C for 48 h, the CF presents low moisture content (6.9%), offering storage stability. The moderate ash content (3.94%) suggests a presence of minerals that can be valuable for health nutrition. The carbohydrates of CF are composed of 13.92% cellulose (glucan content), and 8.21% hemicelluloses (composed by xylan (4.69%) and arabinan (3.52%)). This raw material has a low lipid (fat) content, representing 1.67% of CF total biomass.

Significant levels of aqueous (38.01%) and ethanolic (26.84%) extractives suggest high bioactive and functional molecules availability, which, combined with the higher protein content (22.35%), positions *C. striatus* as a promising feedstock for functional foods, nutraceuticals, and cosmetic products. Moreover, the existent quantities of uronic (4.45%) and acetic (1.01%) acids add the potential of CF for gelling and preservative applications in foods.

The identification and content of principal minerals present in CF were presented in Table 2. The data highlights the nutritional and potential health benefits of these brooms’ flowers due to their essential macro- and microelements. For example, potassium (K, 15,137 mg/kg) is essential for cellular osmoregulation and enzymatic activity; phosphorus (P, 3022 mg/kg), is essential for calcification of bones, and is a cofactor in myriad enzyme systems; calcium (Ca, 2889 mg/kg), is essential during bone formation and beneficial for preventing osteoporosis; magnesium (Mg, 1719 mg/kg) is essential for the functions of many enzyme systems and for neuromuscular transmission; sodium (Na, 845 mg/kg); and other trace elements—Manganese (Mn), Iron (Fe), Zinc (Zn), and Copper (Cu)—with important biological functions, supporting immune system defenses [23,24].

The chemical composition results reported in this work for yellow broom flowers of the species *C. striatus*, collected in the Ourense region of Spain, are comparable with the results presented by Caramelo and colleagues for three flowers from different regions of Portugal [25].

### 2.2. Extraction Conditions Optimization

An experimental design based on a response surface methodology (RSM) was conducted to optimize the extraction conditions for the recovery of antioxidant phenolic compounds. Table 3 summarizes the independent variables studied (ethanol concentration, temperature, and extraction time) and the subsequent dependent variables (TPC, TFC, DPPH, ABTS, FRAP, and CUPRAC). Additionally, Table 4 displays the regression coefficients obtained from a second-degree polynomial equation as follows: the correlation (R^2^) and statistical significance (Fisher’s F test) parameters, and the statistical significance (based on Student’s t-test). The values of R^2^ > 0.92 (in all cases) reflect a positive suitability of the model regarding the real relationship between selected variables, in addition to the high F values (up to 26.12) which imply a good fit to the model.

Significant regression coefficients (higher than 90%) can be employed to calculate the quadratic regression equations of the dependent variables (Y1–Y6):

(Y_1_) $TPC=51.18+2.49x_{1}+6.08x_{2}+5.67x_{1}x_{2}+3.09x_{1}x_{3}+2.67x_{2}x_{3}-3.98x_{1}x_{1}+2.38x_{2}x_{2}.$

(Y_2_) $TFC=96.24+19.07x_{1}+9.32x_{2}+6.58x_{1}x_{2}.$

(Y_3_) $DPPH=143.73+13.49x_{1}+7.48x_{2}+12.35x_{1}x_{2}+10.02x_{1}x_{3}-11.33x_{1}x_{1}.$

(Y_4_) $ABTS=33.83+2.97x_{1}+4.83x_{2}+1.14x_{3}+2.97x_{1}x_{2}+1.64x_{1}x_{3}+1.68x_{2}x_{3}-1.50x_{1}x_{1}+2.03x_{2}x_{2}-1.20x_{3}x_{3}.$

(Y_5_) $FRAP=81.88+3.85x_{1}+11.50x_{2}+4.61x_{1}x_{2}-9.56x_{1}x_{1}.$

(Y_6_) $CUPRAC=540.41+57.53x_{1}+57.04x_{2}+18.56x_{3}+65.88x_{1}x_{2}+28.72x_{1}x_{3}-50.21x_{1}x_{1}.$

#### 2.2.1. Total Phenolic Content (TPC)

The TPC values ranged from 39.03 to 74.74 mg GAE/g CF. As shown in Table 3, the best results were obtained at higher temperatures (150 °C), residence time (9 min) and ethanol concentration (79%), highlighting the maximum value of 74.74 mg GAE/g CF. On the other hand, the lowest ethanol concentration (0%, i.e., 100% water) reached the lowest TPC (39.03 mg GAE/g CF). This behavior evidences the positive effect of ethanol (or ethanol with low quantities of water) as a solvent for the extraction of phenolic compounds due to its affinity for these kinds of molecules [26,27]. This is also observed in Table 3 due to the positive effect of ethanol concentration (x_1_) and the quadratic effect of ethanol (x_1_x_1_), besides its interaction with temperature (x_1_x_2_) and time (x_1_x_3_). The significant effects of ethanol and time were also reported by others when assessing TPC in *Vernonia amygdalina* leaves processed via microwave technology [28].

Figure 2a displays the response surfaces for TPC, exhibiting higher values when employing larger residence times and higher ethanol concentrations. In the same line, Lohvina et al. [29] studied the TPC of ethanolic extracts (30–96% *v*/*v*) of fenugreek (*Trigonella foenum-graecum* L.) leaves and seeds, observing a clear higher phenolic content when employing 70% ethanol in either feedstock. Comparable results were reported by Yusof et al. [30] when processing propolis with ethanol, attaining up to 8.90 mg GAE/mL at 80% ethanol, in comparison with 1.46 mg GAE/mL when using 20% ethanol.

According to Luís et al. [31], phenolic compounds from *Cytisus multiflorus* are better extracted by ethanol than water, presenting TPC values of 120 mg GAE/g CF.

In another study, the authors reported that the highest phenolic content of *Cytisus* flower extract- TPC of 4.9 mg GAE/g, TFC of 14.0 mg QE/g dry extract—was achieved under optimal ultrasound-assisted extraction conditions. These conditions comprised a solvent concentration of 78%, which aligns closely with the concentration revealed in our study, with an extraction temperature of 52 °C during 55 min [32].

The differences observed between the results obtained in this study and those reported in the literature can be attributed to several factors, including the *Cytisus* species and its origin, the specific conditions of the extraction process (such as temperature, type of solvent, extraction time and solid/liquid ratio, and extraction technology), as well as the analytical methods employed for the quantification of phenolic compounds.

#### 2.2.2. Total Flavonoid Content (TFC)

The flavonoid concentration of the extracts varied widely between 67.30 and 150.05 mg RE/g CF, exhibiting a greater affinity for higher ethanol concentration, larger residence time, and higher temperature (Run 10: 79% ethanol, 150 °C, 9 min). However, only the effect of ethanol concentration, temperature of processing, and their interaction were significant, as observed in Table 3. The effect of temperature on TFC was also significant when studying the extraction of antioxidant phenolics from avocado seed by deep eutectic solvents [33], whereas both ethanol concentration and residence time were significant when studying the TFC of ethanolic samples from ultrasonic extraction of avocado peel [34].

Figure 2b, displaying the effect of ethanol and time on the flavonoid content, reflects a similar behavior to TPC (Figure 2a), reaching higher values at greater solvent concentration and residence time. Similarly, the microwave-assisted processing of *Vernonia amygdalina* leaves with 20–80% ethanol produced extracts with higher TFC when employing 80% ethanol, reaching up to 91.12 mg quercetin equivalent/g [28], which may be in the range of those obtained in this study. Results in a similar range were observed by Nguyen et al. [35] when processing *Avicennia officinalis* with different organic solvents such as methanol, acetone, or ethanol.

#### 2.2.3. Antioxidant Activity of Extracts

The antioxidant capacity of the extracts was evaluated by in vitro assays, namely DPPH, ABTS, FRAP, and CUPRAC. In all cases, the ethanol concentration, temperature, and the concomitant effect of ethanol-temperature, and the negative quadratic effect of ethanol concentration had a significant consequence on the results.

Figure 2c represents the variation in DPPH with the residence time and ethanol. In this case, the simultaneous effect of ethanol concentration-time significantly affected the results. The values varied from 99.59 to 182.51 mg TE/g CF, which are in the range of those obtained when processing avocado seed by ethanolic extraction with microwaves [36]. The obtained results demonstrate the positive impact of the highest values of the variables studied (ethanol, temperature, and time) (see Table 3). A similar trend was observed by Ismail-Suhaimy et al. [37], presenting a significant increase in DPPH values when using 80% ethanol rather than higher or lower concentrations, although not showing significant variations when changing the residence time between 30 and 150 s. Conversely, the ethanolic extraction by microwaves on spent filter coffee resulted in higher antioxidant capacity measured by DPPH when using lower ethanol concentration (20%) and shorter reaction times (40 s) at 80 W of power [38].

A similar tendency was observed for ABTS results, displayed in the response surface of Figure 2d. In this case, all the operational conditions presented a significant effect on the results (Table 3). Similarly, results by Niu et al. [39] depicted that ethanol concentrations of 70% enabled maximum ABTS scavenging rates of extracts from *Alpinia oxyphilla* leaves. Another study investigated the influence of ethanol concentration, microwave power, and extraction time on *Barleria lupulina* leaves, finding that higher ethanol concentrations and lower power and extraction times favored higher ABTS values [37].

FRAP values ranged up to 99.77 mg TE/g CF, highlighting a maximum area around ethanol 50% and larger residence times (see Figure 2e). A similar effect was found in the study by Lin et al. [40], reaching a maximum value of FRAP when extracting strawberry leaves with 50% ethanol. Conversely, *Robinia pseudoacacia* wood subjected to hydrothermal extraction by microwaves reached analogous FRAP values (102.30 mg TE/g of wood) [21].

Finally, the CUPRAC response surface (Figure 2f) was similar to those obtained for the other antioxidant capacity assays, reaching the highest value (745.84 mg TE/g CF) at 79% ethanol, temperature of 150 °C and 9 min of residence time. The positive effect of time and ethanol-time interaction was also evident (see Table 3). Kayahan and Saloglu [41] revealed that a concentration of 50% ethanol allowed the greatest CUPRAC values on Turkish artichokes processed by microwaves, highlighting a significant effect of the extraction time.

Several investigations suggest that plants’ antioxidant effect may protect against oxidative stress-related disorders. Flavonoids and other polyphenols are the most studied in this regard. As far as we know, CFs are rich in phenolic acids and flavonoids [15]. Sundararajan et al. [42] found that 104.0 µg/mL *C. scoparius* flower extracts protect around 50% of the rat liver from Fe II/ascorbate system-induced lipid peroxidation. *Cytisus* extracts showed a decrease in hydroxyl radical formation with an IC_50_ value of 27.0 µg/mL, compared to conventional vitamin E (32.5 μg/mL). In addition, González et al. [14] demonstrated that CF extracts present potential for topical application in skin protection, owing to their strong antioxidant capacity against oxidative damage.

#### 2.2.4. Optimized Conditions

The main objective of performing the experimental design on CF was to determine the optimal extraction conditions leading to the highest TPC, TFC, and antioxidant capacity values. For that reason, the software Statgraphics Centurion XVI was selected to evaluate those conditions by converting the response values of each variable using a desirability function in order to obtain the variables that would enable the maximization of the responses for TPC, TFC, DPPH, ABTS, FRAP, and CUPRAC values concomitantly.

Hence, the predicted maximum values would be an ethanol concentration of 87.64%, at a temperature of 160.78 °C, and a residence time of 8.76 min. The optimized conditions were evaluated in triplicate, and the experimental and predicted results are shown in Table 5. Overall, the optimized conditions led to similar values to those that were maximal within the experimental design. The effectiveness of the RSM for quantitative forecasting was validated by the strong correlation between the predicted and actual results. This consistency justified the selection of the experimental design, which demonstrated excellent accuracy and reliability in predicting the TPC, TFC, and antioxidant capacity of the extracts.

### 2.3. Phenolic Compounds of Cytisus Flower Extract

*Cytisus* spp. contains a wide range of phytocompounds belonging to different chemical groups. It is reported that these species are very rich in alkaloids (e.g., sparteine) with anti-arrhythmic properties and bioactive phenolic compounds like genistein and chrysin [43]. Phenolic compounds are secondary metabolites from plants that are present in small amounts in nature. Although they are not synthesized by humans, these compounds play crucial actions in body homeostasis maintenance, particularly in the management of oxidative stress [44].

The results presented in Table 6 revealed that the optimized CF extract exhibits a vast profile of phenolic compounds, including phenolic acids (like chlorogenic acid, protocatechuic acid, and gallic acid), flavonoids (like quercetin, kaempferol, apigenin, and chrysin derivatives), catechol and derivatives (1,2-benzenediol (catechol), 2,5-dihydroxybenzaldehyde, 2,4-dihydroxyacetophenoe), coumarin (esculetin or 6,7 dihydroxycoumarin), phenolic acid glycoside (erigeside C), benzidine compound—erianin (chemically named 2-methoxy-5-[2-(3,4,5-trimethoxy-phenyl)-ethyl]-phenol), and anthraquinone derivative (9-dehydroxyeurotinone). These compounds are known for their health-promoting properties, particularly their antioxidant, anti-inflammatory, anticancer, and antimicrobial activities [26,45,46,47,48].

The intensity values were used to estimate the abundance of these compounds; whereby, higher intensities suggest higher concentrations of the respective phenolic compound in the CF extract.

Chrysin 7-(4″-acetylglucoside) is a flavonoid glucoside known for antioxidant, antidiabetic, and anti-inflammatory properties [49,50,51,52]. Also, 4-Hydroxybenzaldehyde is a well-known antioxidant, anti-angiogenic and anti-nociceptive, anti-inflammatory and antimicrobial activities [53]. The high intensity of these compounds suggests that they are the most abundant phenolic compounds in the *Cytisus* flower extract, playing a key role in its overall bioactivity profile and its pharmacological properties.

Other compounds present in the optimized CF extract also exhibit relevant health applications such as functional foods, nutraceuticals, and pharmaceuticals. For example, esculetin, a coumarin compound with strong antioxidant properties, helps alleviate arthritis, diabetes, cancer, and liver disorders by inhibiting oxidative stress and inflammatory pathways [47].

Quercetin and kaempferol are also key flavonoids commonly found in plant extracts and were notably identified in our optimized CF extract. Studies have demonstrated their beneficial effects in reducing blood pressure and inflammation, with both compounds being effectively absorbed and metabolized in the liver. These findings support the potential cardiovascular benefits of plant-based sources rich in quercetin and kaempferol [54].

Some studies have characterized the phenolic profile of flowers from various *Cytisus* species, revealing a large spectrum of these of bioactive compounds. Studies have highlighted the presence of several classes of phenolic compounds, with flavonoids being the most abundant group. Flavones and flavonols, in particular, are frequently identified in the flowers of *Cytisus* species, contributing significantly to their antioxidant and anti-inflammatory properties [31,32,55,56,57].

Lores et al. [10] analyzed the extracted phenolic compounds from *C. scoparius* using ethyl lactate and pressurized solvent extraction, followed by LC–MS/MS. Their study showed that the main bioactives found in the *Cytisus* extract were the non-flavonoid phenolic compounds caffeic and protocatechuic acids and 3,4-dihydroxybenzaldehyde; the flavonoids rutin, kaempferol, and quercetin; the flavones chrysin, orientin, and apigenin; and the alkaloid lupanine. Larit et al. [57] reported the presence of several flavonoids, also identified in our extract—including genistein, chrysin, chrysin-7-*O*-β-d-glucopyranoside, and 2”-*O*-α-l-rhamnosylorientin—in extracts from aerial parts of *C. villosus*. Pereira and colleagues [56] revealed the phenolic composition of *C. multiflorus*, identifying chrysin-7-*O*-β-d-glucopyranoside and of considerable amounts of a dihydroxyflavone isomer of chrysin are the main phenolic compounds quantified in the extract. Additionally, luteolin derivatives (such as 2″-*O*-pentosyl-6-*C*-hexosyl-luteolin, 2″-*O*-pentosyl-8-*C*-hexosyl-luteolin, and orientin), apigenin and its derivatives, as well as quercetin derivatives, were identified and quantified in the CF extract.

Barros et al. [55], have characterized the phenolic profile of *C. multiflorus* using HPLC–DAD–ESI/MS. The predominant phenolic compounds found in the CF extract include flavonoids, particularly chrysin derivative (35 mg/g dry extract). Other flavonoids such as kaempferol, quercetin, luteolin, and apigenin derivatives were also identified, contributing to the extract overall bioactivity.

Moreover, it is essential to emphasize the synergistic effects of the phenolic compounds in *Cytisus* extract, as well as their interactions with other bioactive components such as polysaccharides, proteins, and minerals. The combination of phenolic compounds in the CF extract likely contributes to a synergistic enhancement of its bioactivity. Numerous studies suggest that phenolic compounds exhibit greater effectiveness when working in concert, rather than in isolation, which may account for the broad-spectrum health benefits attributed to plant extracts. This synergistic action could amplify the anti-inflammatory, antioxidant, and other therapeutic effects of the extract, making *CF* of *striatus* species a valuable source of natural antioxidants with imperative applications as functional foods, cosmetics, and pharmaceutical ingredients.

Furthermore, recent studies have highlighted that the synergistic effects among phenolic compounds not only enhance their antioxidant and anti-inflammatory activities but also influence specific molecular targets and metabolic pathways relevant to human health. These interactions can modulate cellular redox balance and immune regulation. Moreover, phenolic compounds can shape the gut microbiota, promoting the growth of beneficial bacteria and suppressing pathogenic species, and enhancing short-chain fatty acid (SCFA) production/SCFA receptor expression, thereby supporting host metabolic functions and intestinal homeostasis [58,59]. Such effects are particularly relevant in functional food applications, where whole-plant extracts offer advantages over isolated compounds due to their combinatorial bioactivity profiles and microbiome-modulating capacity.

### 2.4. Cytotoxicity of Cytisus Flower Extract

Figure 3 shows the cell viability of two cell lines (HEK293t, human embryonic kidney cells; and A549, human lung carcinoma cells) against increasing concentrations of CF extracts (0–2000 µg/mL). Analyzing the obtained results, both tested cell lines show a slight initial decrease in viability at low CF extract concentrations (0–250 µg/mL). Interestingly, HEK293t showed some viability values above 100% initially, suggesting slight cell proliferation or metabolic stimulation at very low CF extract concentrations. A549 cells exhibited a small drop to around 90% viability, suggesting they are more sensitive to the CF extract even at lower doses.

At medium to high concentrations of CF extract (250–2000 µg/mL), the HEK293t cells viability stabilized around 85–90%, without strong cytotoxic effects even at high concentrations; and A549 cells viability remained lower, around 80–85%, but still above the critical toxicity threshold (commonly considered 60–70%). Based on these results, the CF extracts do not cause major cytotoxicity in either HEK293t or A549 cells up to 2000 µg/mL; however, the extract appears to be relatively safe within the tested concentration range, but A549 cells are slightly more sensitive compared to HEK293t cells.

In addition, as shown in Figure 4f, the cell viability of macrophages used in the anti-inflammatory assay exhibited a slight reduction with *Cytisus* extract treatment (tested at 312, 625, and 1250 µg/mL). At 625 µg/mL of CF extract, the cell viability remained consistently above 70% for all treatment groups, with no significant differences compared to the control group. At 1250 µg/mL, the extract resulted in approximately 58% macrophages viability.

Caramelo and colleagues [25] consider that ethanolic extracts of CF extracts (same *C. striatus* species) do not show toxicity in colon cancer cells (Caco2 cell line) at doses between 50 and 200 ug/mL, showing viability of over 89%. On the other hand, Bouziane et al. [11] evaluated the cytotoxic effects of aqueous and ethyl acetate extracts from aerial parts of *C. villosus* using the MTT assay in two human breast cancer cell lines (T47D, MCF7) and one colon cancer cell line (HCT116). Both extracts inhibited cell proliferation, with ethyl acetate extract IC_50_ values between 1.57 and 3.2 mg/mL and aqueous extract between 2.6 and 5.4 mg/mL (higher dosages than in our study).

These differences may be attributed to variations in *Cytisus* species, extraction solvents, and phytochemical profiles present in the extracts. It is also important to note that additional cytotoxicity tests should be conducted using other cell lines.

These results can open the door for future research, because the slight increase in viability at low concentrations for HEK293t could be interpreted as a hormetic effect (low-dose stimulation, high-dose inhibition) [60], and greater sensitivity of cancer cells compared to normal cells is observed, suggesting selectivity, which could be interesting for therapeutic exploitation (e.g., anticancer properties) [26,61].

### 2.5. Anti-Inflammatory Response of Cytisus Flower Extract

Inflammation is a complex and highly regulated immune response activated by biological (e.g., microorganisms), chemical (e.g., heavy metals and cigarette smoke), or physical agents (e.g., UV radiation and excessive exercise) causing tissue damage or infection. Normally, in physiological conditions, inflammation is a transient and protective response; however, chronic inflammation, characterized by persistent overproduction of cytokines and other inflammatory mediators, can contribute to the development of cancer and other diseases [62].

Given the pivotal role of cytokines in the regulation of inflammation, the present study evaluated the anti-inflammatory action of CF extract by analyzing the modulation of key cytokines, specifically pro-inflammatory cytokines (TNF-α, IL-1β, and IL-6) and dual-role cytokines (IL-10 and TGF-β1) in macrophages. While pro-inflammatory cytokines promote inflammatory pathways, anti-inflammatory cytokines contribute to the treatment or regulation of inflammatory processes [63].

Phenolic compounds, key bioactive ingredients in plant extracts, are well-known to reduce causes involved in the inflammatory process [44]. Phenolic compounds are known for their anti-inflammatory properties, with anthocyanins, flavonoids (e.g., chrysin, quercetin, catechin, rutin, etc.), and some phenolic acids like gallic acid, having demonstrated efficacy in reducing the expression of pro-inflammatory cytokines [51]. The anti-inflammatory activity of phenolic compounds may be due to their ability to balance oxidative stress in cells and suppress the signaling of pro-inflammatory transduction and inhibit the production of inflammatory cytokines [19,44,51].

In this study, *CF* extract, rich in phenolic compounds (see Table 6), was evaluated for its anti-inflammatory effect by measuring cytokine release from non-activated and activated macrophages in vitro (Figure 4).

As reported in the previous Section 3.4, macrophage viability exhibited a slight reduction following treatment with the *Cytisus* extract; however, it remained consistently above 70% across all treatment groups, with no significant differences compared to the control group (Figure 4f).

The CF extract exhibited its most significant effect in LPS-activated macrophages, where it reduced the pro-inflammatory cytokine IL-1β while enhancing IL-10 levels, thereby shifting the cytokine profile towards an anti-inflammatory response (Figure 4).

These findings align with previous studies demonstrating the modulatory effects of phenolic compounds on inflammatory processes in vitro [19].

These results suggest that *C. striatus* flower extract suppresses inflammasome activation induced by LPS. Moreover, a comparison with existing literature reveals a lack of prior data on *Cytisus* flowers, which highlights this study as the first to demonstrate their anti-inflammatory effect.

*Cytisus* flowers are rich in phenolic compounds, particularly the flavonoid chrysin (5,7-dihydroxyflavone), which has attracted considerable attention due to its recognized bioactives, making it a promising natural source of this compound for potential food and therapeutic applications. Some of the literature reviews related to the antioxidant and anti-inflammatory actions of chrysin and several pharmacological activities like anticancer, neuroprotective, antidiabetic, and antiarthritic, among others [51,64].

Del Fabbro and coworkers [49] described that chrysin reduces neuroinflammation by decreasing the expression of important inflammatory enzymes such as inducible nitric oxide synthase (iNOS) and cyclooxygenase-2 (COX-2), downregulating pro-inflammatory cytokines (TNF-α, IL-6, and IL-1β), and blocking NF-κB signaling. Along with improving antioxidant defenses by increasing the activity of antioxidant enzymes like glutathione peroxidase and superoxide dismutase. Moreover, the authors reported that chrysin inhibits apoptosis via controlling the expression of apoptotic markers.

Faheem et al. [65] reported the in vivo action of chrysin in the reduction in the arthritis score, by regulation of inflammatory TNF-α, NF-κB, and TLR-2 levels, and increased the anti-inflammatory cytokines IL-4 and IL-10. Interestingly, it decreased inflammation and joint damage, showing effects comparable to those of the non-steroidal anti-inflammatory drug Piroxicam, indicating its potential as an anti-inflammatory and immunomodulatory agent for the treatment of arthritis.

In another in vivo study, chrysin treatment for 16 weeks in type 2 diabetic rats significantly improved renal function and reduced oxidative stress. It downregulated TNF-α expression, inhibited NF-κB activation, and suppressed TGF-β, fibronectin, and collagen-IV in renal tissues. Additionally, chrysin lowered serum IL-1β and IL-6 levels. These findings suggest that chrysin mitigates diabetic nephropathy via anti-inflammatory effects [52]. Also, Ramírez-Espinosa et al. [50] revealed that chrysin has significant acute antihyperglycemic and antidiabetic effects in nude diabetic mice, impairing the generation of pro-inflammatory cytokines involved in the development of diabetes (IL-1β and TNF-α).

These results consistently demonstrate the broad anti-inflammatory, antioxidant, and immunomodulatory properties of chrysin across different pathological models.

The 4-hydroxybenzaldehyde, another significant compound in our CF extract, appears to be able to inhibit nitric oxide (NO) production and induce COX-2 and iNOS in LPS-activated macrophages. In addition, it has the ability to reduce the level of reactive oxygen species (ROS), showing antioxidant and anti-inflammatory effects [53].

Apigenin is recognized as an effective antioxidant, antimicrobial, and anti-inflammatory natural compound. Patil and colleagues confirm the anti-inflammatory effect of apigenin by inhibiting the LPS-induced expression of iNOS, COX-2, expression of pro-inflammatory cytokines (IL-1β, IL-2, IL-6, IL-8, and TNF-α), and AP-1 proteins and nitric oxide production in LPS-induced inflammation in human lung cells—A549 [66].

These and other compounds present in the CF extract, which may also exhibit anti-inflammatory activities, could act synergistically to further enhance the overall therapeutic effects of the extract.

To conclude the discussion, some limitations of the current study are outlined below to provide context for the interpretation of the results and to guide future research. This study provides relevant insights into the in vitro antioxidant and anti-inflammatory potential of *C. striatus* flower extracts. These assessments may not fully reflect in vivo responses due to differences in metabolism, absorption, and bioavailability. Another important point is the relative toxicity of the extract; although the optimized CF extract showed low cytotoxicity in selected cell lines, a broader panel of normal and cancerous cells would be necessary to comprehensively evaluate safety and selectivity. Additionally, the gut microbiota modulation, synergistic interactions, and mechanistic pathways were inferred from existing literature and not directly tested. Future studies involving animal models, clinical validation, and more advanced metabolomics techniques are needed to confirm and extend these findings.

## 3. Materials and Methods

### 3.1. Chemicals

Folin–Ciocalteu reagent, 2,2′-Azino-bis(3-ethylbenzothiazoline-6-sulfonic acid) diammonium salt (ABTS), 2,2-Di(4-tert-octylphenyl)-1-picrylhydrazyl (DPPH), 2,4,6-Tris(2-pyridyl)-s-triazine (TPTZ), 6-hydroxy-2,5,7,8-tetramethylchroman-2-carboxylic acid (Trolox), aluminum chloride (AlCl_3_), 3-(4,5-dimethylthiazol-2-yl)-2,5-diphenyltetrazolium bromide (MTT), Dulbecco’s Modified Eagle Medium (DMEM), fetal bovine serum (FBS), penicillin-streptomycin solution, resazurin sodium salt, dimethyl sulfoxide (DMSO, ≥99.9%), lipopolysaccharides (LPS), and all standard markers for HPLC were obtained from Sigma-Aldrich (St. Louis, MO, USA). The Mouse Uncoated ELISA kit (Invitrogen) was purchased from Thermo Fisher Scientific Inc. (Waltham, MA, USA). Other reagents were analytical grade, and ultra-pure water was used throughout the experiments.

### 3.2. Chemical Characterization of Plant Material

*Cytisus striatus* flowers (CF) were collected in La Merca, Ourense, Spain (June 2023). Firstly, the flowers were cleaned and dried at 37 °C with aeration for 48 h and subsequently milled in a cutting mill (Retsch SM 2000, Retsch GmbH, Haan, Germany) to a granulometry of 0.45–0.7 mm.

Chemical characterization of CF was accessed in accordance with the National Renewable Energy Laboratory (NREL) official protocols and included aqueous and ethanol extractives (NREL/TP-510-42619), structural carbohydrates (namely cellulose and hemicellulose), acid insoluble residue (NREL/TP-510-42618), and ash content (NREL/TP-510-42622). Uronic acids (galacturonic acid equivalents) were quantified following the Blumenkrantz and Asboe-Hansen method [67]. Fat content was determined according to the official AOAC method (nº 920.39). Total protein content, estimated by using the N×6.25 conversion factor, was performed using a Kjeldahl distillator (Kjeltec 8400 Analyzer, FOSS, Hilleroed, Denmark) by quantification of nitrogen after CF digestion. Moisture was determined gravimetrically using a moisture analyzer (MAC 50/1/NH, RADWAG, Radom, Poland). Mineral content was determined by inductively coupled plasma atomic emission spectrometry (ICP-OES, iCAP PRO XP Duo, Thermo Scientific, Dreieich, Germany) after CF microwave digestion with HNO_3_/HCl/H_2_O_2_.

All experiments were performed at least in triplicate.

### 3.3. Optimization of Extraction Conditions and Extract Preparation

A Monowave 450 single-mode microwave reactor (Anton Paar Spain S.L.U., Madrid, Spain) operating with a power of 850 W, equipped with a temperature sensor mechanism, was employed as equipment for MAE. The extraction of phenolic compounds present in CF was carried out in a Pyrex vessel of 30 mL at a 1:15 solid–liquid ratio (SLR) and heated up to the target temperature as fast as possible. The samples were magnetically stirred (600 rpm), and the temperature was determined using an infrared detector.

Experiments to study the extraction conditions were performed using response surface methodology (RSM, 2^3^ central composite design). The levels of independent variables were selected based on the results obtained from our preliminary experiments and data from the literature. The five levels of each of the three variables were coded in 17 runs (including three replicates of the center point) and were performed in a random order. Independent variables of extraction were EtOH concentration (X_1_, 0 to 98.9% (*v*/*v*)), temperature (X_2_, 98.5 to 163.4 °C), and time (X_3_, 1 to 11 min). Dependent variables (Y_1_ to Y_6_) were total phenolic content (TPC, mg GAE/g CF), total flavonoid content (TFC, mg CE/g CF), 2,2-Diphenyl-1-picrylhydrazyl radical scavenging assay (DPPH, mg TE/g CF), 2,2′-azino-bis(3-ethylbenzothiazoline-6-sulfonic acid) radical cation-based assay (ABTS, mg TE/g CF), FRAP ferric reducing antioxidant power (FRAP, mg TE/g CF), and cupric reducing antioxidant capacity assay (CUPRAC, mg TE/g CF), respectively. Coded and real values of the independent variables and results of dependent variables are presented in Table 1. Data were correlated following the polynomial Equation (1).(1)$$Y_{i}=\beta_{0i}+\beta_{1i}x_{1}+\beta_{2i}x_{2}+\beta_{3i}x_{3}+\beta_{11i}x_{1}^{2}+\beta_{22i}x_{2}^{2}+\beta_{33i}x_{3}^{2}+\beta_{12i}x_{1}x_{2}\;+\beta_{13i}x_{1}x_{3}+\beta_{23i}x_{2}x_{3}$$
where Y_i_ corresponds to the dependent variables; *x*_1_, *x*_2_ and *x*_3_ value of independent variables; β_0*i*_, β_1*i*_, β_2*i*_, β_3*i*_, β_11*i*_, β_22*i*_, β_33*i*_, β_12*i*_, β_13*i*_, and β_23*i*_ are regression coefficients calculated from experimental data by multiple regression using the least-squares method.

Experimental data were fitted by applying the regression analysis function of Microsoft Excel’s Data Analysis Add-In, USA. The adequacy of the model was demonstrated by appraising the coefficient of determination (R^2^), the significance of the regression coefficients, and the F-test value obtained from the analysis of variance.

The optimization was performed using the desirability function of the software Statgraphics Centurion version XVI (Statpoint Technologies Inc., Warrenton, VA, USA). The model was confirmed experimentally by three independent assays under the optimal extraction conditions.

After each extraction, the liquid extracts were separated by vacuum filtration (11 µm) and stored at −20 °C until further analysis. The extract obtained under the optimum conditions was repeated several times and lyophilized to have enough material for subsequent tests.

### 3.4. HPLC-TOF-MS Analyses of the Extracts

A tentative identification of CF phenolic compounds recovered under optimal extraction conditions was carried out by liquid chromatography coupled with trapped ion mobility spectrometry and TOF high-resolution mass spectrometry (HPLC-TOF-MS). The sample was injected into a ZORBAX Eclipse XDB-C18 rapid resolution HD (2.1 × 100 mm 1.8 Micron, Agilent Technologies, Santa Clara, CA, USA) and LC separation was carried out on an Elute HPLC (Bruker Daltonics). The mobile phases employed were as follows: 0.1% aqueous formic acid (solvent A) and 0.1% formic acid in acetonitrile (solvent B) at a 0.4 mL/min flow rate. The linear gradient was as follows: 2% solvent B over 2 min, from 2% to 30% solvent B over 13 min, 30% to 100% solvent B over 2 min, 100% solvent B over 4 min, 100% to 2% solvent B over 1 min, and then isocratically 2% solvent B for 2 min. Ions were generated by an ESI source in negative and positive ion mode. Working conditions were as follows: 4500 V capillary voltage, 500 V end plate offset, 8.0 L/min dry N_2_ gas, 8 bar nebulizer pressure, and 240 °C dry heater. Identification of metabolites was based on the smart formula, the accurate mass data, isotopic pattern matching (*m*Sigma value), retention time, and compounds reported in the literature.

### 3.5. Antioxidant Activity

Four different methods to assess the antioxidant capacity of the bioactive molecules of CF extracts were used, i.e., 2,2-diphenyl-β-picrylhydrazyl radical scavenging assay (DPPH), 2,2-azino-bis-3-ethylbenzothiazoline-6-sulfonic acid radical cation decolorization assay (ABTS), ferric reducing antioxidant power (FRAP), and cupric reducing antioxidant capacity (CUPRAC). DPPH, ABTS, and FRAP were performed as previously described methods [68] and CUPRAC [69].

The 6-hydroxy-2,5,7,8-tetramethylchromen-2-carboxylic acid (Trolox) reagent was used as a standard for all tests, and the results were represented as mg of Trolox equivalents (TE)/g of dry CF.

### 3.6. Cell Viability

The in vitro metabolic activity of optimized CF extract (resuspended in DMSO) was evaluated using human embryonic kidney (HEK293T, ATCC^®^ CRL-11268, passage 15) human lung adenocarcinoma cells (A549, ATCC^®^ HTB-38™, passage 9). These cell lines were kindly provided by Dr. Andreia Gomes (Department of Biology, University of Minho).

Cell metabolic activity was assessed using the resazurin reduction assay [70]. Cells were cultured in Dulbecco’s Modified Eagle Medium (DMEM) supplemented with 10% fetal bovine serum (FBS) and 1% penicillin/streptomycin and maintained at 37 °C in a humidified atmosphere containing 5% CO_2_. Once cells reached 70–80% confluence, they were trypsinized and seeded into 96-well plates at a density of 1 × 10^5^ cells/mL.

Cells were incubated with supplemented DMEM containing the extracts and controls (without extract with or without 0.5% DMSO) at concentrations ranging from 0 to 2000 µg/mL for 24 h. Following incubation, the culture medium was replaced with 200 µL of fresh medium containing resazurin (0.5 mM in PBS). After a 2 h incubation at 37 °C, 150 µL of the supernatant was transferred to a new 96-well plate, and fluorescence from the reduced product (resorufin) was measured at 560 nm excitation and 590 nm emission wavelengths using a microplate reader.

Each experimental condition was performed in triplicate. Cell viability (%) was calculated after subtracting blank values (cell-free medium) and expressed relative to untreated control cells (0.5% DMSO).

### 3.7. Anti-Inflammatory Activity

The macrophage cell line (J774A.1, ATCC TIB-67 ™) was cultured in DMEM high glucose, supplemented with 10% FBS, 2 mM glutamine, 1 mM sodium pyruvate, and 25 mM HEPES buffer. The cultures were maintained in tissue culture flasks (Nagle Nunc, Int., Hereford, UK) with a humidified atmosphere containing 5% CO_2_ at 37 °C (Binder CB150; Tuttlingen, Germany). After confluent growth, macrophage cells were washed with a fresh medium and recovered by scraping. Viable cells were counted by Trypan blue exclusion in the hemocytometer and resuspended in DMEM to a final concentration of 1 × 10^6^ cells/mL. A volume of 300 µL of the macrophage suspension was then cultured in 48-well tissue culture plates.

Cells were incubated overnight with or without 1 µg/mL of lipopolysaccharides (LPS, Merk) to establish inflammation models, respectively [19]. Following incubation, cells were washed with fresh medium and treated with 625 µg/mL of CF extract for 24 h. The concentration of extract was selected based on preliminary cell viability results (Section 2.4) that demonstrated no significant cytotoxicity in the MTT assay. After treatment, the supernatants were collected and stored at −20 °C for cytokine quantification.

The metabolic viability of the cells was determined using the MTT assay [71]. Formazan crystals formed were dissolved in a DMSO:ethanol (1:1) solution, and absorbance was measured at 570 nm. Controls included cells incubated only with DMEM, with the extracts’ solvent (PBS), and with 1 µg/mL of LPS.

The concentrations of Tumor Necrosis Factor (TNF)-α, interleukin (IL)-1β, IL-6, IL-10, and transforming growth factor (TGF)-β1 in cell culture supernatants were measured using the corresponding Mouse Uncoated ELISA kit (Invitrogen by Thermo Fisher Scientific, Waltham, MA, USA), following the manufacturer’s instructions.

Each experiment was performed in three independent experiments, and each in triplicate.

### 3.8. Statistical Analysis

All experiments were carried out at least in triplicate, and the data are presented as mean ± standard deviation (SD) values. GraphPad Prism^®^ software (version 8.0; San Diego, CA, USA) was used for statistical analyses. The analysis of variance (ANOVA) and Tukey’s multiple comparisons test were used to determine statistically different values at a significance level of *p* < 0.05.

## 4. Conclusions

This study highlights the potential of *Cytisus striatus* flowers using food-grade solvents (water and ethanol) associated with an innovative extraction method—MAE. Extraction process optimization successfully enhanced the recovery of antioxidant and anti-inflammatory phenolic compounds from *C. striatus* flower, under optimized conditions of 87.6% ethanol, 160.8 °C, and 8.76 min. The optimized extract showed high levels of phenolic and flavonoid content, containing 27 identified phenolic compounds, including abundant chrysin derivatives, apigenin, and quercetin, distinguished for their antioxidant activity. The extract exhibited low cytotoxicity on both normal (HEK293t) and cancerous (A549) cells up to 2000 µg/mL, maintaining viability above 80%, and safe profiles were also confirmed in macrophages. Anti-inflammatory effects were demonstrated by a remarkable reduction in pro-inflammatory cytokines (particularly IL-1β) and an increase in anti-inflammatory cytokine IL-10 levels in LPS-activated macrophages.

This study revealed that *C. striatus* flower extracts could be promising candidates for use in functional foods, nutraceuticals, and cosmetic applications, in line with the principles of the circular economy and the sustainable valorization of resources.

## Figures and Tables

**Figure 1 ijms-26-07100-f001:**
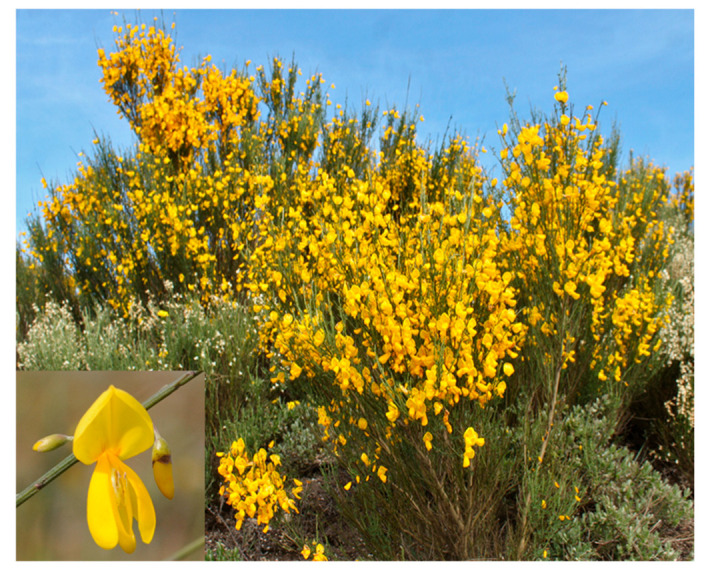
*Cytisus striatus* in full bloom.

**Figure 2 ijms-26-07100-f002:**
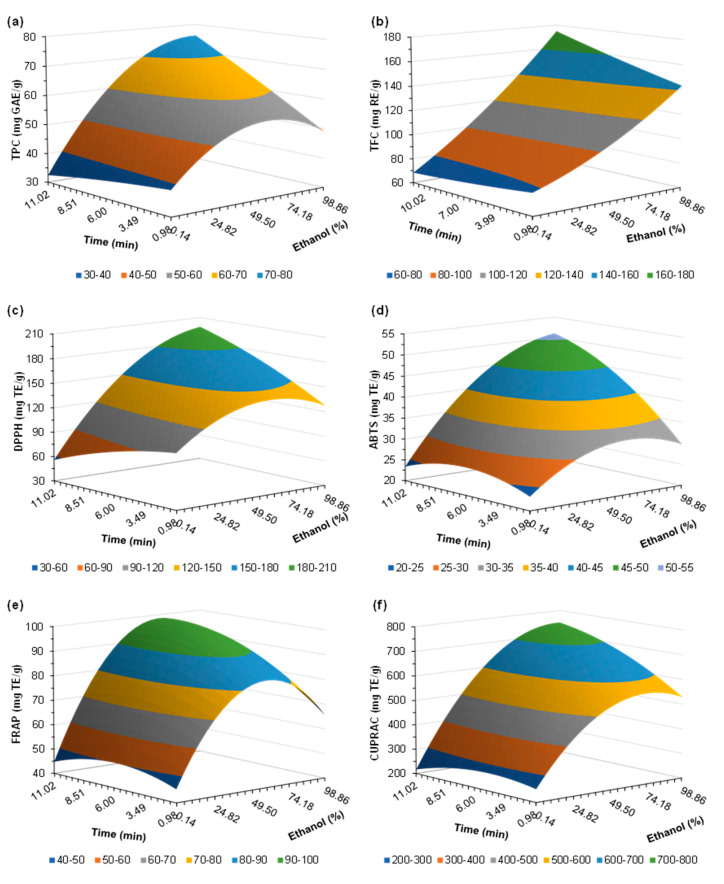
Response surface for (**a**) total phenolic content-TPC; (**b**) total flavonoid content-TFC; (**c**) DPPH; (**d**) ABTS; (**e**) FRAP; and (**f**) CUPRAC, as a function of time and ethanol concentration, fixing the variable temperature to the maximum value (T = 150 °C).

**Figure 3 ijms-26-07100-f003:**
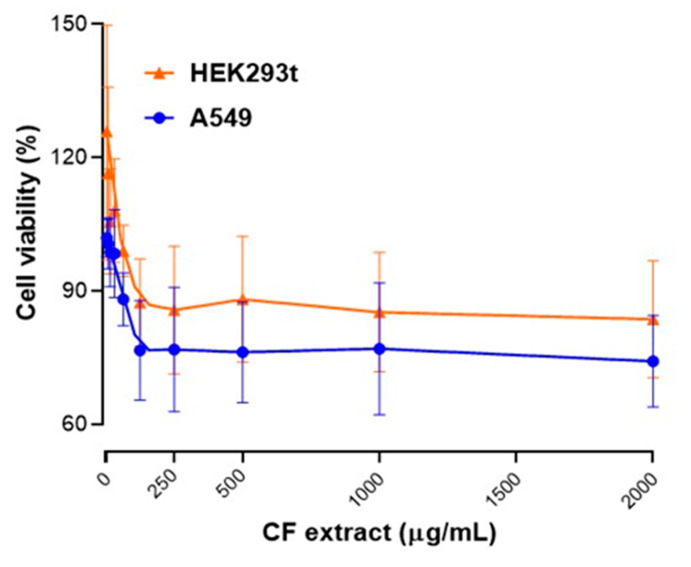
Cellular viability of treated cells with different concentrations of *Cytisus* Flower (CF) extract. Each experiment was performed in five independent experiments, and each in triplicate.

**Figure 4 ijms-26-07100-f004:**
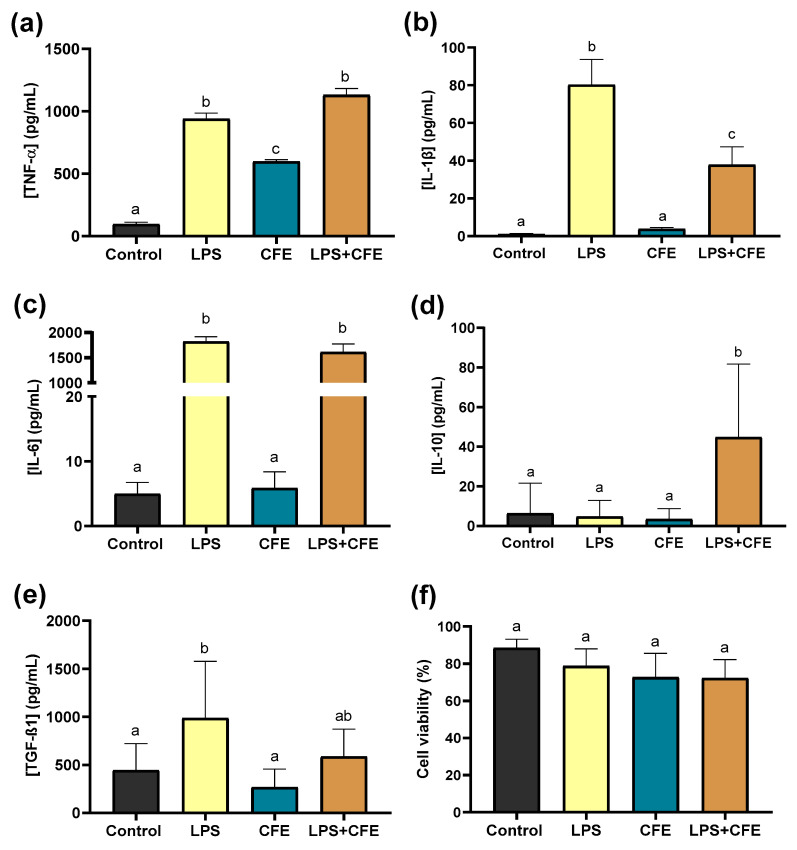
Measurements of cytokines in macrophages after incubation with LPS and *Cytisus* extract (CFE, 625 µg/mL) for 24 h. (**a**) Tumor necrosis factor-α (TNF-α), (**b**) interleukin-1β (IL-1β), (**c**) interleukin-6 (IL-6), (**d**) interleukin-10 (IL-10), and (**e**) transforming Growth Factor-Beta 1 (TGF-β1), (**f**) cell viability. Different letters correspond to a statistical difference between treatment groups. The control group corresponds to 0.5% DMSO vehicle-treated cells. Each experiment was performed in three independent experiments, and each in triplicate.

**Table 1 ijms-26-07100-t001:** Chemical characterization of *Cytisus striatus* flowers (g/100 g of CF).

Component	Mean	SD
Moisture	6.9	0.14
Ash	3.94	0.11
AIR	6.22	0.70
Glucan	13.92	1.69
Xylan	4.69	1.04
Arabinan	3.52	0.44
Acetic acid	1.01	0.11
Uronic acid	4.45	0.93
Aqueous extractives	38.01	0.1
Ethanolic extractives	26.84	0.02
Protein	22.35	0.45
Fat	1.67	0.01

AIR: acid insoluble residue. SD: Standard deviation. All experiments were performed in triplicate.

**Table 2 ijms-26-07100-t002:** Mineral composition of *Cytisus striatus* flowers (mg/Kg of CF).

Minerals	LOQ	Mean	SD
K	12.50	15,136.6	202.4
P	2.50	3021.7	2.2
Ca	2.50	2888.9	95.8
Mg	5.00	1719.0	28.1
Na	12.50	844.6	10.1
Mn	0.25	477.6	4.9
Fe	1.00	64.9	0.9
Zn	1.00	35.1	0.2
Cu	1.00	7.7	0.2

LOQ: Quantification limit. SD: Standard deviation. All experiments were performed in triplicate.

**Table 3 ijms-26-07100-t003:** Studied operational conditions (independent variables) expressed as coded and real values (X_1_, EtOH concentration; X_2_, temperature; X_3_, time) and extraction results reached for dependent variables (Y_1_, TPC; Y_2_, TFC; Y_3_, DPPH; Y_4_, ABTS; Y_5_, FRAP and Y_6_, CUPRAC).

R	X_1_-EtOH (%)	X_2_-T (°C)	X_3_-t (min)	Y_1_-TPC (mgGAE/g CF)	Y_2_-TFC (mg RE/g CF)	Y_3_-DPPH (mg TE/g CF)	Y_4_-ABTS (mg TE/g CF)	Y_5_-FRAP (mg TE/g CF)	Y_6_-CUPRAC (mg TE/g CF)
1	−1 (20)	−1 (110)	−1 (3)	49.27	78.68	123.28	29.24	57.06	437.78
2	1 (79)	1 (150)	−1 (3)	57.37	128.72	145.71	37.64	83.41	587.17
3	0 (49.5)	0 (130)	0 (6)	51.61	97.03	143.70	33.17	82.02	552.92
4	1 (79)	−1 (110)	1 (9)	40.37	103.33	138.91	28.43	54.16	429.40
5	−1 (20)	1 (150)	1 (9)	45.04	87.01	104.38	31.82	71.48	413.73
6	1 (79)	−1 (110)	−1 (3)	43.94	100.69	111.94	27.03	60.67	391.50
7	−1 (20)	1 (150)	−1 (3)	50.28	93.31	114.46	30.26	76.33	444.70
8	0 (49.5)	0 (130)	0 (6)	52.36	96.22	140.42	35.06	82.48	547.09
9	−1 (20)	−1 (110)	1 (9)	43.61	79.49	116.98	26.38	60.25	435.59
10	1 (79)	1 (150)	1 (9)	74.74	150.05	182.51	48.05	98.80	745.84
11	−1.67 (0)	0 (130)	0 (6)	39.03	67.30	99.59	25.73	49.57	300.41
12	1.67 (98.9)	0 (130)	0 (6)	42.39	136.03	137.56	35.86	61.80	515.75
13	0 (49.5)	−1.67 (98.5)	0 (6)	48.84	94.93	148.74	31.98	64.83	504.09
14	0 (49.5)	0 (130)	0 (6)	52.71	95.88	144.45	32.79	80.91	517.94
15	0 (49.5)	1.67 (163.4)	0 (6)	68.20	112.81	178.98	49.34	99.77	670.54
16	0 (49.5)	0 (130)	−1.67 (1)	51.34	94.09	141.43	30.15	72.87	486.97
17	0 (49.5)	0 (130)	1.67 (11)	51.01	98.11	143.19	33.12	83.13	540.16

**Table 4 ijms-26-07100-t004:** Regression coefficients and statistical parameters measuring the correlation and significance of the models.

Coefficient	Y_1_-TPC	Y_2_-TFC	Y_3_-DPPH	Y_4_-ABTS	Y_5_-FRAP	Y_6_-CUPRAC
x_0_	52.18 ^a^	96.24 ^a^	143.73 ^a^	33.83 ^a^	81.88 ^a^	540.41 ^a^
x_1_	2.49 ^b^	19.07 ^a^	13.49 ^a^	2.97 ^a^	3.85 ^b^	57.53 ^a^
x_2_	6.08 ^a^	9.32 ^a^	7.84 ^b^	4.83 ^a^	11.50 ^a^	57.04 ^a^
x_3_	0.17	1.86	3.70	1.14 ^b^	1.80	18.56 ^b^
x_12_	5.67 ^a^	6.58 ^b^	12.35 ^a^	2.97 ^a^	4.61 ^b^	65.88 ^a^
x_13_	3.09 ^b^	3.68	10.02 ^b^	1.64 ^b^	1.32	28.72 ^b^
x_23_	2.67 ^c^	1.45	0.76	1.68 ^b^	1.73	11.50
x_11_	−3.98 ^a^	2.30	−11.33 ^a^	−1.50 ^b^	−9.56 ^a^	−50.21 ^a^
x_22_	2.38 ^b^	3.09	4.84	2.03 ^a^	−0.06	13.81
x_33_	−0.24	0.31	−2.86	−1.20 ^b^	−1.59	−12.53
R^2^	0.945	0.962	0.926	0.971	0.959	0.971
F-exp	13.23	19.55	9.80	25.94	18.29	26.12
Significance level (%)	99.87	99.96	99.67	99.99	99.95	99.99

^a^ Significant coefficients at the 99% confidence level. ^b^ Significant coefficients at the 95% confidence level. ^c^ Significant coefficients at the 90% confidence level.

**Table 5 ijms-26-07100-t005:** Predicted and experimental values under optimum conditions (ethanol, 87.64%; T, 160.78 °C; t, 8.76 min) based on multiple responses of TPC, TFC, DPPH, ABTS, FRAP, and CUPRAC.

	Y_1_-TPC	Y_2_-TFC	Y_3_-DPPH	Y_4_-ABTS	Y_5_-FRAP	Y_6_-CUPRAC
Predicted value	82.46	168.02	203.52	57.57	101.88	838.85
Experimental value	85.9 ± 1.8	120.3 ± 8.0	260.1 ± 12.7	62.9 ± 2.2	105.1 ± 3.5	907.3 ± 9.6

All experiments were performed in triplicate. Data are presented as mean ± standard deviation (SD) values.

**Table 6 ijms-26-07100-t006:** Phenolic compounds tentatively identified in the extract of *Cytisus striatus* flowers obtained under optimized microwave-assisted extraction conditions, as determined by HPLC-TOF-MS. Data include retention time (RT), measured *m*/*z*, mass accuracy deviation (Δ*m*/*z*), type of ion detected, and peak intensity (relative abundance).

Tentative Name	CAS IDs	Molecular Formula	RT [min]	*m*/*z* Meas.	Δ*m*/*z* [ppm]	M Meas.	Ions	mSigma	Peak Intensity
Clorogenic acid	327-97-9	C16H18O9	0.82	353.08779	0.36	354.09507	[M-H]-	20.7	1097
Erigeside C	112667-09-1	C15H20O10	0.81	359.0988	1.198	360.10608	[M-H]-	24.5	1316
Pyrogallol	87-66-1	C6H6O3	0.83	125.02401	−3.061	126.03129	[M-H]-	7.2	31,854
Quercetin 3-galactoside	482-36-0	C21H20O12	0.86	463.08848	0.127	464.09575	[M-H]-	28.2	13,145
3,4-Dihydroxybenzoic acid	99-50-3	C7H6O4	0.87	153.0187	−4.201	154.02598	[M-H]-	6.9	120,796
Esculetin	305-01-1	C9H6O4	0.88	177.01891	−2.373	178.02619	[M-H]-	7.3	248,808
1,2-Benzenediol	120-80-9	C6H6O2	0.89	109.02938	−0.851	110.03666	[M-H]-	2.5	63,753
2,5-Dihydroxybenzaldehyde	1194-98-5	C7H6O3	0.91	137.02388	−3.952	138.03115	[M-H]-	5.0	131,780
2,4-Dihydroxyacetophenone	89-84-9	C8H8O3	0.92	151.03947	−3.981	152.04674	[M-H]-	5.8	19,259
Gallic acid	149-91-7	C7H6O5	1	169.01376	−2.882	170.02104	[M-H]-	9.2	5246
4-Hydroxybenzaldehyde	123-08-0	C7H6O2	3.72	121.02904	−3.845	122.03632	[M-H]-	30.3	1,568,340
trans-o-Coumaric acid 2-glucoside	614-60-8	C15H18O8	6.43	325.09302	0.959	326.10029	[M-H]-	17.7	60,382
Rutin	153-18-4	C27H30O16	9.12	609.1453	−1.24	610.15258	[M-H]-	29.3	17,286
2,5-Dihydroxybenzaldehyde	1194-98-5	C7H6O3	9.17	137.02383	−4.283	138.03111	[M-H]-	0.8	1,312,766
Apigenin-7-glucoside	578-74-5	C21H20O10	9.23	431.09847	−0.301	432.10575	[M-H]-	13.6	232,042
Quercetin 3-galactoside	482-36-0	C21H20O12	9.26	463.08816	−0.167	464.09543	[M-H]-	7.0	116,170
2,5-Dihydroxybenzaldehyde	1194-98-5	C7H6O3	9.81	137.02383	−4.35	138.03111	[M-H]-	2.5	666,716
2-Cinnamoyl-1-galloylglucose	56994-83-3	C22H22O11	9.86	461.10907	0.208	462.11635	[M-H]-	4.7	199,241
Kaempferol	520-18-3	C15H10O6	10.56	285.04042	−0.143	286.0477	[M-H]-	17.6	664,234
Quercetin	6151-25-3	C15H10O7	10.63	301.03529	−0.215	302.04257	[M-H]-	3.2	97,502
9-Dehydroxyeurotinone	1360606-85-4	C15H12O5	11.3	271.06124	0.175	272.06852	[M-H]-	15.2	19,319
Apigenin	520-36-5	C15H10O5	11.31	269.0455	−0.543	270.05278	[M-H]-	13.4	736,030
Chrysin 7-(4″-acetylglucoside)	674299-89-9	C23H22O10	11.56	459.12858	0.001	458.1213	[M+H]+	1.5	16,413,959
Amentoflavone	1617-53-4	C30H18O10	11.93	537.0826	0.245	538.08987	[M-H]-	15.8	11,780
Oroxylin A	480-11-5	C16H12O5	13.05	283.06126	−0.494	284.06854	[M-H]-	17.9	42,832
Erianin	95041-90-0	C18H22O5	14.7	317.13942	2.126	318.1467	[M-H]-	7.3	13,100

## Data Availability

Dataset available on request from the authors.

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
