# Peer review of "Development of Cytisus Flower Extracts with Antioxidant and Anti-Inflammatory Properties for Nutraceutical and Food Uses"

_ijms, 2025, doi:10.3390/ijms26157100_

Round 1
Reviewer 1 Report
Comments and Suggestions for Authors
The authors extracted and identified the phenolic composition of Cytisus striatus flowers using an ultrasound-assisted method. The yield was significantly enhanced through a response surface methodology, demonstrating a green and convenient approach. Additionally, the toxicity and bioactivity of the extracts were evaluated, indicating good application potential. It is recommended to accept the manuscript after addressing the following issues.
Major suggestions:
- Considering oral intake, intestinal and hepatic cells are typically most exposed to bioactive compounds. The rationale for selecting HEK293T (kidney) and A549 (lung) cells for cytotoxicity evaluation should be clarified.
- It is unclear whether phenolic compositions vary among leaves, flowers, and roots/rhizomes of Cytisus species. Please explain why flowers were chosen for extraction and characterization. If flowers possess superior pharmacological properties, stating this would strengthen the manuscript's logic.
- As the study centres on phenolic profiling, the inclusion of nutritional and mineral analysis appears tangential. Clarifying its relevance to phenolic identification and antioxidant assessment would improve the manuscript’s focus.
- Were the ranges of independent variables used in the response surface methodology determined through prior single-factor tests? The use of 160 °C heating raises concerns about potential thermal degradation and oxidation of phenolic compounds, which should be cncerned.
- The authors used a combination of water, ethanol, and ultrasound-assisted extraction, which is common for phenolic extraction. Although the identification of compounds is interesting, the study seems to emphasize mainly the most abundant components. It is suggested that the authors focus on novel or unique compounds and conduct a deeper analysis of the LC-MS data to highlight innovative findings.
- The observation that cell viability exceeds 150% at low extract concentrations is unusual and likely questionable. Additionally, the error bars in the cytotoxicity assays are quite large, with intra-group variability reaching up to 20%. Further evaluation of the cytotoxicity experimental reproducibility is recommended.
- While the authors mention potential applications in functional foods, nutraceuticals, and cosmetics, no specific innovative formulations or product development strategies are provided. This aspect could be elaborated to strengthen the practical relevance of the work. Furthermore, it is recommended to expand the discussion by exploring the synergistic effects among phenolic compounds, their molecular targets, and the impact on gut microbiota in food applications. The following references offer valuable insights and could be cited to support this discussion.DOI: 10.26599/FSHW.2024.9250081; 10.1021/acs.jafc.4c10158.
Formatting suggestions:
- In Tables 1 and 2, the authors report means and standard deviations (SD) but do not specify the number of replicates. Likewise, several figure captions omit essential details regarding the number of experimental repeats. Including this information would enhance clarity and reproducibility.
- It is suggested to add the corresponding CAS Registry Numbers (CAS IDs) for each compound listed in Table 6.
Author Response
REVIEWER #1
The authors extracted and identified the phenolic composition of Cytisus striatus flowers using an ultrasound-assisted method. The yield was significantly enhanced through a response surface methodology, demonstrating a green and convenient approach. Additionally, the toxicity and bioactivity of the extracts were evaluated, indicating good application potential. It is recommended to accept the manuscript after addressing the following issues.
Major suggestions:
- Considering oral intake, intestinal and hepatic cells are typically most exposed to bioactive compounds. The rationale for selecting HEK293T (kidney) and A549 (lung) cells for cytotoxicity evaluation should be clarified.
R: Thank you for this important comment. HEK293T and A549 cells were chosen for the initial toxicity screening because they are well-established human cell lines commonly used in biomedical studies. Although not directly involved in oral absorption, these cell lines provide a general perspective on cytotoxicity in different tissue types. We agree that gastrointestinal and hepatic models would provide more targeted insights, and we intend to incorporate such models in future studies.
- It is unclear whether phenolic compositions vary among leaves, flowers, and roots/rhizomes of Cytisus species. Please explain why flowers were chosen for extraction and characterization. If flowers possess superior pharmacological properties, stating this would strengthen the manuscript's logic.
R: Flowers of Cytisus striatus were selected because they are traditionally used in herbal preparations and infusions. Moreover, previous studies in related species have shown that flowers are particularly rich in bioactive phenolic compounds, often more so than leaves or stems. Their accessibility and high phytochemical content justified their selection for this work.
- As the study centres on phenolic profiling, the inclusion of nutritional and mineral analysis appears tangential. Clarifying its relevance to phenolic identification and antioxidant assessment would improve the manuscript’s focus.
R: We appreciate this observation. Although the study focuses on phenolic profiling, we included nutritional and mineral content analysis to provide a more comprehensive characterization of the flower matrix. This is particularly relevant when considering potential food, nutraceutical, or cosmetic applications, where a broader nutritional profile can enhance product value and safety assessment.
- Were the ranges of independent variables used in the response surface methodology determined through prior single-factor tests? The use of 160 °C heating raises concerns about potential thermal degradation and oxidation of phenolic compounds, which should be concerned.
R: The ranges for the response surface methodology (RSM) were selected based on preliminary single-factor experiments and literature on microwaves-assisted extraction. While 160 °C might seem high, it refers to the microwaves temperature setting, and really sample heating is moderated due to the presence of pressure and short extraction time.
- The authors used a combination of water, ethanol, and ultrasound-assisted extraction, which is common for phenolic extraction. Although the identification of compounds is interesting, the study seems to emphasize mainly the most abundant components. It is suggested that the authors focus on novel or unique compounds and conduct a deeper analysis of the LC-MS data to highlight innovative findings.
R: Thanks for the suggestion. We have revised the discussion section to highlight the phenolic compounds identified in the extract, with important documented bioactive impact. While abundant compounds are usually highlighted due to higher quantification, we agree that the new findings add scientific value and have extended our analysis accordingly.
“Other compounds present in the optimized CF extract also exhibit relevant health applications as functional foods, nutraceuticals and pharmaceuticals. For example, esculetin, a coumarin compound with strong antioxidant properties, helps alleviate arthritis, diabetes, cancer, and liver disorders by inhibiting oxidative stress and inflammatory pathways [47].
Quercetin and kaempferol are also key flavonoids commonly found in plant extracts, and were notably identified in our optimized CF extract. Studies have demonstrated their beneficial effects in reducing blood pressure and inflammation, with both compounds being effectively absorbed and metabolized in the liver. These findings support the potential cardiovascular benefits of plant-based sources rich in quercetin and kaempferol [54].”
- The observation that cell viability exceeds 150% at low extract concentrations is unusual and likely questionable. Additionally, the error bars in the cytotoxicity assays are quite large, with intra-group variability reaching up to 20%. Further evaluation of the cytotoxicity experimental reproducibility is recommended.
R: The observation of cell viability exceeding 120% (not 150% has mentioned by the reviewer) likely reflects a proliferative or metabolic stimulation effect at low extract concentrations. However, we acknowledge error bars in the cytotoxicity assays are quite large, and for this reason, in this assay, we performed 5 independent experiments to confirm reproducibility of data.
- While the authors mention potential applications in functional foods, nutraceuticals, and cosmetics, no specific innovative formulations or product development strategies are provided. This aspect could be elaborated to strengthen the practical relevance of the work. Furthermore, it is recommended to expand the discussion by exploring the synergistic effects among phenolic compounds, their molecular targets, and the impact on gut microbiota in food applications. The following references offer valuable insights and could be cited to support this discussion.DOI: 10.26599/FSHW.2024.9250081; 10.1021/acs.jafc.4c10158.
R: We have expanded the discussion to include more detail on the potential practical applications of the extract, particularly in functional foods and cosmetic formulations. Additionally, we now address the possible synergistic effects among phenolic compounds, their molecular mechanisms, and implications for gut microbiota. The recommended references have been incorporated to support these points.
“Furthermore, recent studies have highlighted that the synergistic effects among phenolic compounds not only enhance their antioxidant and anti-inflammatory activities but also influence specific molecular targets and metabolic pathways relevant to human health. These interactions can modulate cellular redox balance and immune regulation. Moreover, phenolic compounds can shape the gut microbiota, promoting the growth of beneficial bacteria and suppressing pathogenic species, and enhancing short-chain fatty acid (SCFA) production/SCFA receptor expression, thereby supporting host metabolic functions and intestinal homeostasis [57,58]. Such effects are particularly relevant in functional food applications, where whole-plant extracts offer advantages over isolated compounds due to their combinatorial bioactivity profiles and microbiome-modulating capacity.”
Formatting suggestions:
- In Tables 1 and 2, the authors report means and standard deviations (SD) but do not specify the number of replicates. Likewise, several figure captions omit essential details regarding the number of experimental repeats. Including this information would enhance clarity and reproducibility.
R: Thank you. This information is included in the manuscript.
“All experiments were carried out at least in triplicate, and the data are presented as mean ± standard deviation (SD) values”
“Each experiment was performed in five independent experiments, and each in triplicate.”
- It is suggested to add the corresponding CAS Registry Numbers (CAS IDs) for each compound listed in Table 6.
R: Thank you for the suggestion. The CAS IDs has been included in table 6 as suggested.
Reviewer 2 Report
Comments and Suggestions for Authors
1. The choice of flower species and relevant literature must be clearly justified and elaborated in the introduction section. Please revise lines 61–67 accordingly, providing a strong rationale for selecting the Cytisus flower.
2. An image or photograph of the Cytisus flower should be included to enhance reader understanding.
3. At line 345, there is a spelling error. A comprehensive spell check of the entire manuscript is recommended to identify and correct similar issues.
4. The standard deviations presented in Figure 2 are notably high and should be carefully reviewed. Please clarify the possible reasons or correct the data as necessary.
5. Lines 386–398 are overly general and do not contribute meaningfully to the discussion. These lines should be removed.
6. Western blot data supporting Figure 3 is currently missing. Please include the Western blot images to support the findings.
7. The conclusion should be presented as a single paragraph.
8. The limitations of the current study must be clearly stated in the revised manuscript.
Author Response
REVIEWER #2
- The choice of flower species and relevant literature must be clearly justified and elaborated in the introduction section. Please revise lines 61–67 accordingly, providing a strong rationale for selecting the Cytisus flower.
R: Thank you for the comment. We have revised the Introduction to provide a clearer justification for choosing Cytisus striatus flowers, supported by relevant literature on their traditional use and phytochemical richness.
- An image or photograph of the Cytisus flower should be included to enhance reader understanding.
R: An image of Cytisus striatus flowers has been added to the revised manuscript to assist readers in visually identifying the species.
Figure 1. Cytisus striatus in full bloom.
- At line 345, there is a spelling error. A comprehensive spell check of the entire manuscript is recommended to identify and correct similar issues.
R: The entire manuscript has been spell-checked to correct possible errors.
- The standard deviations presented in Figure 2 are notably high and should be carefully reviewed. Please clarify the possible reasons or correct the data as necessary.
R: We reviewed the original data. The high standard deviations were due to the natural biological variability of this type of experiment. Therefore, in these experiments we conducted in five independent experiments, each one in triplicate, in order to ensure the accuracy of the data. These tests showed that cell viability is not significantly affected by CF extracts.
- Lines 386–398 are overly general and do not contribute meaningfully to the discussion. These lines should be removed.
R: Thank you for your suggestion. These sentences have been shortened to better contribute to the discussion.
- Western blot data supporting Figure 3 is currently missing. Please include the Western blot images to support the findings.
R: Thank you for the comment. The concentrations of Tumour Necrosis Factor (TNF)-α, interleukin (IL)-1β, IL-6, IL-10, and Transforming growth factor (TGF)-β1 in cell culture supernatants were measured using the corresponding Mouse Uncoated ELISA kit (Invitrogen), following the manufacturer’s instructions. For this reason, western blots were not carried out in this study.
- The conclusion should be presented as a single paragraph.
R: The conclusion section has been presented into a single, as requested by the reviewer.
- The limitations of the current study must be clearly stated in the revised manuscript.
R: A dedicated paragraph outlining the limitations of this study has been added to the end of the discussion section.
“To conclude the discussion, some limitations of the current study are outlined below to provide context for the interpretation of the results and to guide future research. This study provides relevant insights into the in vitro antioxidant and anti-inflammatory potential of C. striatus flower ex-tracts. These assessments may not fully reflect in vivo responses due to differences in metabolism, absorption, and bioavailability. Another important point is relatively the toxicity of extract; although the optimized CF extract showed low cytotoxicity in selected cell lines, a broader panel of normal and cancerous cells would be necessary to comprehensively evaluate safety and selectivity. Additionally, the gut microbiota modulation, synergistic interactions, and mechanistic pathways were inferred from existing literature and not directly tested. Future studies involving animal models, clinical validation, and more advanced metabolomics techniques are needed to con-firm and extend these findings.”
Round 2
Reviewer 2 Report
Comments and Suggestions for Authors
The manuscript is now improved significantly. There are no more comments. Hence, it is recommended for publication.
Author Response
Thank you for taking the time to review this manuscript. This will improve the quality of the document.